# On the optimization and generalization of overparameterized implicit neural networks

## Abstract

Implicit neural networks have become increasingly attractive in the machine learning community since they can achieve competitive performance but use much less computational resources. Recently, a line of theoretical works established the global convergences for first-order methods such as gradient descent if the implicit networks are over-parameterized. However, as they train all layers together, their analyses are equivalent to only studying the evolution of the output layer. It is unclear how the implicit layer contributes to the training. Thus, in this paper, we restrict ourselves to only training the implicit layer. We show that global convergence is guaranteed, even if only the implicit layer is trained. On the other hand, the theoretical understanding of when and how the training performance of an implicit neural network can be generalized to unseen data is still under-explored. Although this problem has been studied in standard feed-forward networks, the case of implicit neural networks is still intriguing since implicit networks theoretically have infinitely many layers. Therefore, this paper investigates the generalization error for implicit neural networks. Specifically, we study the generalization of an implicit network activated by the ReLU function over random initialization. We provide a generalization bound that is initialization sensitive. As a result, we show that gradient flow with proper random initialization can train a sufficient over-parameterized implicit network to achieve arbitrarily small generalization errors.

## 1 Introduction

Implicit neural networks El Ghaoui et al. (2021) have been renewed interest in the machine learning community recently as it can achieve competitive or dominated performance in many applications compared to traditional neural networks while using significantly less memory resources Bai et al. (2019); Dabre & Fujita (2019). Unlike traditional neural networks, feature vectors in implicit layers are not provided recursively. Instead, they are solutions to an equilibrium equation induced by implicit layers. Consequently, an implicit neural network is equivalent to an *infinite-depth* neural network with weight-tied and input-injected. Thus, the gradients can be computed through implicit differentiation Bai et al. (2019) using constant memory.

Empirical success of implicit neural networks has been observed in a number of applications such as natural language processing Bai et al. (2019), compute vision Bai et al. (2022), optimization Ramzi et al. (2021), and time series analysis Rubanova et al. (2019). However, the theoretical understanding of implicit neural networks is still limited compared to conventional neural networks. One of the essential questions in the deep learning community is whether a simple first-order method can converge to a global minimum. This question is even more significant and complicated in the case of implicit neural networks. Since the network has infinitely many layers, it may not be well-posed. Specifically, the equilibrium equation may admit zero or multiple solutions, so the forward propagation is probably not well-posed or even divergent. Many works in the literature Chen et al. (2018); Bai et al. (2019; 2021); Kawaguchi (2021) observe instability of forwarding pass along training epochs: the number of iterations the forward pass uses to find the equilibrium point grows with training epochs. Thus, a line of works put efforts into dealing with this well-posedness issue Winston & Kolter (2020); El Ghaoui et al. (2021); Xie et al. (2022); Gao et al. (2021). Then some recent studies successfully show global convergence of gradient flow and gradient descent for implicit networks. For example, Kawaguchi (2021) proves the convergence of gradient flow for

a linear implicit network, but its result cannot be applied to nonlinear activation. Gao et al. (2021) obtains the global convergence results for ReLU-activated implicit networks if the width of networks is quadratic of the sample size. However, the output layer in their setup is combined with the feature vector, so their results can only be applied in a restricted range of applications. The output issue is solved in their following work Gao & Gao (2022), and the network size is reduced to linear widths. Since they train all layers together, it can be considered a perturbed version of only training the output layer. It is hard to justify the contribution of the implicit layer to the training process. In addition, their work cannot be directly applied to generalization analysis which is another essential problem in the machine learning community.

One essential mystery in the deep learning community is that the neural networks used in practice are often heavily overparameterized such that they can even fit random labels, while they can achieve small generalization errors (*i.e.*, test, error). Although this problem has been studied extensively in standard feed-forward networks Arora et al. (2019); Cao & Gu (2020); Allen-Zhu et al. (2019); Cao & Gu (2019); Jacot et al. (2018), the case of implicit models is still intriguing because implicit networks have infinitely many layers. Unfortunately, there is no study on the generalization theory for learning an implicit neural network to our best knowledge. As more and more successes of implicit networks are observed in practice, there is increasing demand for theoretical analysis to support these observations. In this paper, we initiate the exploration of generalization errors for implicit neural networks. We study one main class implicit network called deep equilibrium models Bai et al. (2019) that is activated by the ReLU activation function. By coupling the implicit network with a kernel machine, we can show that the implicit neural network trained by random initialized gradient flow can achieve arbitrarily small generalization errors if the implicit network is sufficiently overparameterized. Moreover, the generalization bound we obtained in this paper is initialization sensitive. This type of generalization error itself also contributes to the deep learning community. It justifies the observation that initialization is essential for a network to generalize. Consequently, it supports some start-of-the-techniques, such as pre-training, to achieve better test performance. In addition, it could provide a more accurate estimate of the test performance in practice.

**Main contribution**: In this paper, we conduct analyses on the optimization and generalization of an implicit neural network activated by the ReLU activation function.

- We train the implicit neural network through random initialized gradient flow. If the neural network is overparameterized, then gradient flow converges to a global minimum at a linear rate (with high probability). Although this result is obtained in some previous works Gao et al. (2021); Gao & Gao (2022), our fine-gained analysis and proof method set our work apart from the previous works as we only train the implicit layer and the convergence result can be further used in the generalization analysis.

- We couple the implicit neural network with a kernel machine. We use Rademacher complexity theory to provide an initialization-sensitive generalization bound for an overparameterized implicit neural network. As a result, the generalization error of this implicit neural network can be reduced to arbitrarily small if the width of the implicit layer is sufficiently large.

- Some concrete examples of random initialization are provided to show that the assumptions made in the previous contributions can easily be satisfied. Under these specified random initializations, we provide another generalization bound independent of initialization. This generalization bound is easy to compute and independent of the size of the neural network.

## 2 PRELIMINARIES OF IMPLICIT DEEP LEARNING

**Notation**: We use $\|\boldsymbol{x}\|$ to denote its Euclidean norm of a vector $\boldsymbol{x}$ and $\|\boldsymbol{A}\|$ to denote the operator norm of a matrix $\boldsymbol{A}$. For a square matrix $\boldsymbol{A}$, $\lambda_{\min}(\boldsymbol{A})$ denote the smallest eigenvalue of $\boldsymbol{A}$. We use $\text{vec}(\boldsymbol{A})$ to denote the vectorization operation applied on the matrix $\boldsymbol{A}$. Given a function $Y = f(X)$, the derivative $\partial f / \partial X$ is defined by $\text{vec}(dY) = (\partial f / \partial X)^T dX$, where $X$ and $Y$ can be either scalars, vectors, or matrices. We also denote $[n] := [1, 2, \cdots, n]$ to simplify our notation.

In this paper, we consider a main class of implicit networks called deep equilibrium model with $m$ neurons in the implicit layer defined by

$$f_\theta(\boldsymbol{x}) := \frac{1}{\sqrt{m}} \boldsymbol{z}^T \boldsymbol{b}, \tag{1}$$

$$\boldsymbol{z} := \lim_{\ell \to \infty} \boldsymbol{z}^\ell = \lim_{\ell \to \infty} \sigma(\gamma \boldsymbol{A}^T \boldsymbol{z}^{\ell-1} + \phi(\boldsymbol{x})), \tag{2}$$

$$\phi(\boldsymbol{x}) := \phi(\boldsymbol{W}^T \boldsymbol{x}), \tag{3}$$

where $\boldsymbol{x} \in \mathbb{R}^d$ is the input, $\boldsymbol{z} \in \mathbb{R}^m$ is an equilibrium point of the transition equation 2, $\boldsymbol{b} \in \mathbb{R}^m$ is the weight vector in the output layer, $\boldsymbol{A} \in \mathbb{R}^{m \times m}$ is the weight matrix shared among implicit layers, $\boldsymbol{W} \in \mathbb{R}^{d \times m}$ is the weight matrix in the first layer, and $\sigma(z) = \max\{0, z\}$ is the ReLU activation function and $\phi$ is another activation function that is *not* necessary to be ReLU, but we assume $\phi$ is also 1-Lipschitz continuous for simplicity. For convince, we denote $\boldsymbol{\theta} := \text{vec}\,(\boldsymbol{W}, \boldsymbol{A}, \boldsymbol{b})$ as the parameter vector. Here the scalar $\gamma \in (0, 1)$ is a fixed scalar that ensures the transition equation 2 is well posed (Gao & Gao, 2022, Lemma 3.1.).

We are given $n$ input-output samples $S := \{(\boldsymbol{x}_i, y_i)\}_{i=1}^n$ that are drawn i.i.d. from some underlying distribution $\mathcal{D}$. We denote $\boldsymbol{X} := [\boldsymbol{x}_1, \ldots, \boldsymbol{x}_n] \in \mathbb{R}^{n \times d}$, $\boldsymbol{y} := (y_1, \ldots, y_n) \in \mathbb{R}^n$, $\boldsymbol{\Phi} := [\boldsymbol{\phi}_1, \ldots, \boldsymbol{\phi}_n] \in \mathbb{R}^{n \times m}$, and $\boldsymbol{Z} := [\boldsymbol{z}_1, \ldots, \boldsymbol{z}_n] \in \mathbb{R}^{n \times m}$. For simplicity, we further assume for each sample $\|\boldsymbol{x}\| = 1$ and $|y| \le 1$.

We train this implicit network through randomly initialized gradient flow (GF) on the square loss over the sample $S$. Specifically, we initialize the parameters as follows

$$b_r \overset{i.i.d.}{\sim} \mathcal{U}\{-1, +1\}, \quad \boldsymbol{a}_r \overset{i.i.d.}{\sim} subG(\boldsymbol{0}, \boldsymbol{I}_m), \quad \boldsymbol{w}_r \overset{i.i.d.}{\sim} subG(\boldsymbol{0}, \boldsymbol{I}_d), \quad \forall r \in [m]. \tag{4}$$

where $\mathcal{U}$ is the discrete uniform distribution with probability $1/2$, $subG$ is some sub-Gaussian distribution with zero mean and unit variance. We fix the first layer $\boldsymbol{W}$ and output layer $\boldsymbol{b}$ and only optimize the implicit layer $\boldsymbol{A}$ through gradient flow on the objective function as follows

$$L(\boldsymbol{A}) := \frac{1}{2} \|\boldsymbol{u} - \boldsymbol{y}\|^2, \tag{5}$$

where we use $\boldsymbol{u} := f_\theta(\boldsymbol{X})$ to simplify notation.

Suppose the forward propagation is well-posed. Then equilibrium point $\boldsymbol{Z}$ is well defined. By using the implicit function theorem, we can write the gradient flow update as

$$\frac{d\text{vec}\,(\boldsymbol{A})}{dt} = -\frac{\partial L}{\partial \boldsymbol{A}} = -\frac{\gamma}{\sqrt{m}} \left[ \boldsymbol{D}(\boldsymbol{I}_m \otimes \boldsymbol{Z}) \right]^T \boldsymbol{Q}^{-T} \left( \boldsymbol{b}^T \otimes \boldsymbol{I}_n \right)^T (\boldsymbol{u} - \boldsymbol{y}), \tag{6}$$

where $\boldsymbol{D} := \mathbf{diag}[\text{vec}\,(\sigma'(\gamma \boldsymbol{Z}\boldsymbol{A} + \boldsymbol{\Phi}))]$ and $\boldsymbol{Q} := \boldsymbol{I}_{Nm} - \gamma \boldsymbol{D}(\boldsymbol{A}^T \otimes \boldsymbol{I}_n)$.

Let $(\boldsymbol{x}, y)$ be an unseen data following the underlying distribution $\mathcal{D}$. Then the essential learning task is to find a mapping $f$ for which it minimizes the *generalization error* or *population risk*

$$R(f) := \mathbb{E}_{(\boldsymbol{x}, y) \sim \mathcal{D}} \left[ \ell(f(\boldsymbol{x}), y) \right],$$

where $\ell$ is a selected loss function.

## 3  OPTIMIZATION

This section presents our convergence result for random initialized gradient flow. The analysis of gradient flow is a stepping stone toward understanding the discrete algorithms. A line of recent works for discrete algorithms such as gradient descent and stochastic gradient descent is based on or inspired by the analysis of gradient flow Arora et al. (2018); Du et al. (2018); Gao et al. (2021).

Assume $\boldsymbol{Z}(t)$ is well defined at time step $t$. By using a simple chain rule and equation 6, then dynamics of $\boldsymbol{u}(t)$ under gradient flow can be written as follows

$$\frac{d\boldsymbol{u}}{dt} = \left( \frac{\partial \boldsymbol{u}}{\partial \boldsymbol{A}} \right)^T \left( \frac{d\text{vec}\,(\boldsymbol{A})}{dt} \right) = -\boldsymbol{H}(t)(\boldsymbol{u}(t) - \boldsymbol{y}), \tag{7}$$

where $\boldsymbol{H}(t) \in \mathbb{R}^{n \times n}$ and $\boldsymbol{J}(t) \in \mathbb{R}^{n \times mn}$ are defined by

$$\boldsymbol{H}(t) := \left(\frac{\partial \boldsymbol{u}}{\partial \boldsymbol{A}}\right)^T \left(\frac{\partial \boldsymbol{u}}{\partial \boldsymbol{A}}\right) = \frac{\gamma^2}{m} \boldsymbol{J}(t) \left[\boldsymbol{I}_m \otimes \boldsymbol{Z}(t)\boldsymbol{Z}(t)^T\right] \boldsymbol{J}(t)^T, \tag{8}$$

$$\boldsymbol{J}(t) := \left[\boldsymbol{b}^T \otimes \boldsymbol{I}_n\right] \boldsymbol{Q}(t)^{-1} \boldsymbol{D}(t). \tag{9}$$

Clearly, $\boldsymbol{H}(t)$ is a positive semi-definite matrix (PSD). If the smallest eigenvalue $\boldsymbol{H}(t)$ is lower bounded by some positive constant $\lambda_0 > 0$, then solving the ordinary differential equation (ODE) above immediately implies $L(t) := L(\boldsymbol{A}(t))$ converges to 0 at an exponentially rate, *i.e.*, $L(t) \le e^{-\lambda_0 t} L(0)$. It is indeed the case for overparameterized implicit neural networks. Suppose $\lambda_{\min}[\boldsymbol{H}(0)]$ is lower bounded. Then we can show $\boldsymbol{H}(t)$ is close to $\boldsymbol{H}(0)$ so that $\lambda_{\min}[\boldsymbol{H}(t)]$ is also lower bounded if the implicit network is sufficiently overparameterized.

On the other hand, one needs to ensure that forward propagation is well-posed throughout the training. It follows from (Vershynin, 2018, Theorem 4.4.5) that $\|\boldsymbol{A}(0)\| = \mathcal{O}\left(\sqrt{m}\right)$ with high probability. Thus, by choosing $\gamma := \gamma_0/\sqrt{m}$ for small $\gamma_0 \in (0, 1)$, the forward propagation becomes a contraction mapping and the equilibrium point $\boldsymbol{Z}(0)$ is uniquely determined. Although $\boldsymbol{A}(t)$ are varied during the training, we can show $\|\boldsymbol{A}(t)\| = \mathcal{O}\left(\sqrt{m}\right)$ is always the case throughout the training. Thus, $\boldsymbol{Z}(t)$ is always well defined.

**Theorem 3.1** *For any $\delta \in (0, 1)$, choose $m = \Omega\left(\frac{N^4}{\lambda_0^4 \delta}\right)$ and $\gamma = \gamma_0/\sqrt{m}$ for a fixed $\gamma_0 \in (0, 1)$. Assume $\lambda_{\min}(\boldsymbol{H}(0)) \ge \lambda_0$ for some constant $\lambda_0 > 0$ with probability 1 over the random initialization equation 4. Then with probability at least $1 - \delta$ over the random initialization equation 4:*

- $\|\boldsymbol{u}(0) - \boldsymbol{y}\| = \mathcal{O}\left(\sqrt{N/\delta}\right)$

- $\boldsymbol{A}(t) = \mathcal{O}\left(\sqrt{m}\right)$

- $\|\boldsymbol{u}(t) - \boldsymbol{y}\|^2 \le e^{-\lambda_0 t}\|\boldsymbol{u}(0) - \boldsymbol{y}\|^2$

### 3.1 PROOF SKETCH OF THEOREM 3.1

Several previous works Gao & Gao (2022); Gao et al. (2021) have successfully established the global convergence results. However, as they train three layers together, the Gram matrix $\boldsymbol{H}(t)$ is a sum of three PSD matrices induced by each layer. Then they only focus their analyses on the evolution of the output layer instead of the implicit layer. This case is similar to solving a perturbed convex optimization problem. Instead, here we only train the implicit layer while the rest is fixed at their initialization. The corresponding Gram matrix is much more complicated. More fine-grained analysis is required to bound $\boldsymbol{H}(t) - \boldsymbol{H}(0)$.

To bound the difference $\boldsymbol{H}(t)$ to its initialization, it is significant to characterize the trajectory of $\boldsymbol{Z}(t)$ and $\boldsymbol{J}(t)$ during training. Specifically, we derive the partial derivatives of $\boldsymbol{Z}$ and $\boldsymbol{J}$ with respect to $\boldsymbol{A}$ as follows.

**Lemma 3.2** *Suppose $\|\boldsymbol{A}\| \le M$ and $\gamma := \gamma_0/M$ for $\gamma_0 \in (0, 1)$. Then the equilibrium point $\boldsymbol{Z}$ is uniquely determined. Moreover, the partial derivatives of $\boldsymbol{Z}$ and $\boldsymbol{J}$ with respect to $\boldsymbol{A}$ are given by*

$$\frac{\partial \boldsymbol{Z}}{\partial \boldsymbol{A}} = \gamma \left[\boldsymbol{D}(\boldsymbol{I}_m \otimes \boldsymbol{Z})\right]^T \boldsymbol{Q}^{-T}, \tag{10}$$

$$\frac{\partial \boldsymbol{J}^T}{\partial \boldsymbol{A}} = \gamma \left[(\boldsymbol{D}\boldsymbol{U}^{-1}\boldsymbol{B})^T \otimes \boldsymbol{D}\boldsymbol{U}^{-1}\right]^T \tag{11}$$

*where we denote $\boldsymbol{U} := \boldsymbol{Q}^T$ and $\boldsymbol{B} := \boldsymbol{b} \otimes \boldsymbol{I}_n$ to simplify derivation.*

Now we are ready to present our proof sketch. Here we only list the results that are also useful for later generalization analysis. Please see the whole proof in Appendix C.

We assume the results hold for all $0 \le s \le t$:

- $\boldsymbol{A}(s) = \mathcal{O}\left(\sqrt{m}\right)$

- $\|\boldsymbol{u}(s) - \boldsymbol{y}\|^2 \leq e^{-\lambda_0 t}\|\boldsymbol{u}(0) - \boldsymbol{y}\|^2$

The gradient $\partial L/\partial \boldsymbol{A}$ can be bounded as follows

$$\|\partial L(s)/\partial \boldsymbol{A}\| \leq c\|\boldsymbol{X}\|_F\|\boldsymbol{u}(s) - \boldsymbol{y}\|.$$

where the scalar $c$ absorbs constants and quantities related to $\gamma_0$, and we use the facts $\|\boldsymbol{W}(0)\| = \mathcal{O}\left(\sqrt{m}\right)$ and $\|\boldsymbol{b}(0)\| = \sqrt{m}$. Then we can bound $\boldsymbol{A}(t) - \boldsymbol{A}(0)$:

$$\|\boldsymbol{A}(t) - \boldsymbol{A}(0)\|_F \leq \int_0^t \|\partial L/\partial \boldsymbol{A}(s)\|\, ds \leq \frac{c}{\lambda_0}\|\boldsymbol{X}\|_F\|\boldsymbol{u}(0) - \boldsymbol{y}\| \leq \sqrt{m},$$

where the last inequality is because $m = \Omega\left(\frac{N^4}{\lambda_0^4 \delta}\right)$. As a result, we have

$$\|\boldsymbol{A}(t)\| \leq \|\boldsymbol{A}(t) - \boldsymbol{A}(0)\|_F + \|\boldsymbol{A}(0)\| = \mathcal{O}\left(\sqrt{m}\right).$$

By Lemma 3.2, we can bound $\boldsymbol{Z}(t)$ and $\boldsymbol{J}(t)$ to their initialization as follows

$$\|\boldsymbol{Z}(t) - \boldsymbol{Z}(0)\|_F \leq \int_0^t \left\|\frac{d\mathrm{vec}\left(\boldsymbol{Z}\right)}{ds}\right\| ds \leq \frac{c}{\lambda_0}\|\boldsymbol{X}\|_F^2\|\boldsymbol{u}(0) - \boldsymbol{y}\|. \tag{12}$$

$$\|\boldsymbol{J}(t) - \boldsymbol{J}(0)\|_F \leq \int_0^t \left\|\frac{d\mathrm{vec}\left(\boldsymbol{J}^T\right)}{ds}\right\| ds \leq \frac{c}{\lambda_0}\|\boldsymbol{X}\|_F\|\boldsymbol{u}(0) - \boldsymbol{y}\|. \tag{13}$$

Moreover, we have for all $0 \leq s \leq t$

$$\|\boldsymbol{J}(s)\| = \left\|\left(\boldsymbol{b}^T \otimes \boldsymbol{I}_n\right)\boldsymbol{Q}(s)^{-1}\boldsymbol{D}(s)\right\| = \mathcal{O}\left(\sqrt{m}\right),. \tag{14}$$

Combining equation 12, equation 13, equation 14, and Lemma A.5, we have

$$\|\boldsymbol{H}(t) - \boldsymbol{H}(0)\| \leq \frac{c\|\boldsymbol{X}\|_F^3\|\boldsymbol{u}(0) - \boldsymbol{y}\|}{\lambda_0\sqrt{m}} \leq \frac{cn^2}{\lambda_0\sqrt{\delta}\sqrt{m}}, \tag{15}$$

where the last inequality is due to $\|\boldsymbol{X}\| = \sqrt{n}$ and $\|\boldsymbol{u}(0) - \boldsymbol{y}\| = \mathcal{O}\left(\sqrt{n}/\sqrt{\delta}\right)$. It follows from $m = \Omega\left(\frac{n^4}{\lambda_0^4 \delta}\right)$ that $\|\boldsymbol{H}(t) - \boldsymbol{H}(0)\| \leq \lambda_0/2$. By Weyl's inequality, we have

$$\lambda_{\min}\left[\boldsymbol{H}(t)\right] \geq \lambda_{\min}\left[\boldsymbol{H}(0)\right] - \|\boldsymbol{H}(t) - \boldsymbol{H}(0)\| \geq \lambda_0/2.$$

Therefore, the dynamics of loss function $L(t)$ satisfies

$$\frac{d}{dt}\|\boldsymbol{u} - \boldsymbol{y}\|^2 = 2(\boldsymbol{u} - \boldsymbol{y})^T\frac{d\boldsymbol{u}}{dt} = -2(\boldsymbol{u} - \boldsymbol{y})^T\boldsymbol{H}(t)(\boldsymbol{u} - \boldsymbol{y}) \leq -\lambda_0\|\boldsymbol{u} - \boldsymbol{y}\|^2.$$

By solving the ordinary differential equation above, we have

$$\|\boldsymbol{u}(t) - \boldsymbol{y}\|^2 \leq e^{-\lambda_0 t}\|\boldsymbol{u}(0) - \boldsymbol{y}\|^2.$$

## 4 GENERALIZATION

In this section, we study the generalization ability of implicit networks trained by randomly initialized gradient flow. Let $f_t(\boldsymbol{x}) := f_{\boldsymbol{\theta}(t)}(\boldsymbol{x})$ be the corresponding implicit neural network at time $t$. Our result is based on the observation made in Domingos (2020), that is, $f_t(\boldsymbol{x})$ is equivalent to a kernel machine whose kernel function is induced by the gradients along the training.

Specifically, let $(\boldsymbol{x}_0, y_0)$ be a data point that could be either training data or unseen data. Its prediction at time $t$ is given by $f_t(\boldsymbol{x}_0)$. Then its dynamics can be written as

$$\frac{df_t(\boldsymbol{x}_0)}{dt} = \left(\frac{\partial f_t(\boldsymbol{x}_0)}{\partial \boldsymbol{A}}\right)^T\left(-\frac{\partial L}{\partial \boldsymbol{A}}\right) = k_t(\boldsymbol{x}_0, \boldsymbol{X})(\boldsymbol{y} - \boldsymbol{u}(t)),$$

where we define a kernel function

$$k_t(\boldsymbol{x}, \boldsymbol{x}') := \left\langle \frac{\partial f_t(\boldsymbol{x})}{\partial \boldsymbol{A}}, \frac{\partial f_t(\boldsymbol{x}')}{\partial \boldsymbol{A}} \right\rangle, \qquad (16)$$

and the mapping $\boldsymbol{x} \mapsto \partial f_t(\boldsymbol{x})/\partial \boldsymbol{A}$ is the corresponding feature map. Taking integration upto time $t$, the prediction of $f_t(\boldsymbol{x}_0)$ can be written as follows

$$f_t(\boldsymbol{x}_0) = f_0(\boldsymbol{x}_0) + \int_0^t k_s(\boldsymbol{x}_0, \boldsymbol{X})(\boldsymbol{y} - \boldsymbol{u}(s))ds. \qquad (17)$$

Here the kernel function $k_t$ changes along the training process, but by applying some simple algebraic tricks, one can show $f_t$ is indeed a kernel machine whose kernel is called *path kernel* introduced in Domingos (2020). However, its Rademacher complexity is hard to compute, and its generalization cannot be determined simply.

Instead, the main idea in this paper is to couple the trajectory of $f_t$ with another function $\hat{f}_t$ whose dynamics are given by

$$\frac{d\hat{f}_t(\boldsymbol{x}_0)}{dt} = k_0(\boldsymbol{x}_0, \boldsymbol{X})(\boldsymbol{y} - \hat{\boldsymbol{u}}(t)),$$

where $\hat{\boldsymbol{u}}(t)$ is the corresponding estimate for training data $(\boldsymbol{X}, \boldsymbol{y})$ with dynamics given by

$$\frac{d\hat{\boldsymbol{u}}}{dt} = -\boldsymbol{H}(0)(\hat{\boldsymbol{u}}(t) - \boldsymbol{y}). \qquad (18)$$

Then the prediction of $\hat{f}_t$ for data $(\boldsymbol{x}_0, y_0)$ is given by

$$\hat{f}_t(\boldsymbol{x}_0) = f_0(\boldsymbol{x}_0) + \int_0^t k_0(\boldsymbol{x}_0, \boldsymbol{X})(\boldsymbol{y} - \hat{\boldsymbol{u}}(s))ds, \qquad (19)$$

where we use the fact $\hat{f}_0(\boldsymbol{x}_0) = f_0(\boldsymbol{x}_0)$.

We provide a generalization bound in the following theorem for any Lipschitz continuous loss function $\ell$ by showing $f_t$ is close to $\hat{f}_t$ if the implicit network is sufficiently overparameterized.

**Theorem 4.1** *Fix $\delta \in (0, 1)$. Suppose $m = \Omega(\lambda_0^{-8}\delta^{-1}N^{10})$ and $\gamma = \gamma_0/\sqrt{m}$ for some $\gamma_0 \in (0, 1)$. Assume $\lambda_{\min}[\boldsymbol{H}(0)] \geq \lambda_0 > 0$ with probability $1$. Then with probability at least $1 - \delta$ over the random initialization and random training sample S, the implicit neural network $f_\infty$ trained by gradient flow has generalization error $R(f_\infty) := \mathbb{E}[\ell(f_\infty(\boldsymbol{x}), y)]$ bounded by*

$$R(f_\infty) \leq \mathcal{O}\left( \sqrt{\frac{(\boldsymbol{y} - \boldsymbol{u}(0))^T \boldsymbol{H}(0)^{-1}(\boldsymbol{y} - \boldsymbol{u}(0))}{n}} + \sqrt{\frac{\log(1/\delta)}{n}} \right). \qquad (20)$$

The dominating term in equation 20 is :

$$\sqrt{\frac{(\boldsymbol{y} - \boldsymbol{u}(0))^T \boldsymbol{H}(0)^{-1}(\boldsymbol{y} - \boldsymbol{u}(0))}{n}}.$$

This can be used to predict the test accuracy of the learned neural network. It is worth noting that our bound identifies the sensitivity of initialization. If one has a good start, say $\|\boldsymbol{y} - \boldsymbol{u}(0)\|$ is small, then a better test accuracy could probably be achieved. Moreover, let $\boldsymbol{H}(0) = \boldsymbol{Q}\boldsymbol{\Lambda}\boldsymbol{Q}^T$ be the eigenvalue decomposition of $\boldsymbol{H}(0)$. Then the dominating term can be written as

$$\sqrt{\sum_{i=1}^n \lambda_i^{-2}(\boldsymbol{q}_i^T(\boldsymbol{y} - \boldsymbol{u}(0)))^2/n} \leq \sum_{i=1}^n \lambda_i^{-1} \left|\boldsymbol{q}_i^T(\boldsymbol{y} - \boldsymbol{u}(0))\right|/\sqrt{n}$$

$$\leq \left(\sum_{i=1}^n \lambda_i^{-1}\right) \|\boldsymbol{y} - \boldsymbol{u}(0)\|/\sqrt{n}.$$

To have better test performance, we may want to have $\boldsymbol{y} - \boldsymbol{u}(0)$ to align the eigenspace spanned by the eigenvectors with large eigenvalues. Note that $\sum_{i=1}^n \lambda_i = \|\boldsymbol{H}(0)\|_F = \mathcal{O}(n)$. Thus, the test performance can be further improved if eigenvalues $\lambda_i$ are close to each other, which probably explains why gradient descent has implicit regularization induced from discretization. We will consider that as our future work.

## 4.1 PROOF SKETCH OF THEOREM 4.1

Before showing $f_t$ is close to $\hat{f}_t$, we first provide some essential properties of $\hat{f}_t$.

**Lemma 4.2** *Assume $\lambda_{\min}[\boldsymbol{H}(0)] \geq \lambda_0 > 0$. Then the function $\hat{f}_t$ has the following properties for all $t \geq 0$:*

- $\boldsymbol{y} - \hat{\boldsymbol{u}}(t) = e^{-\boldsymbol{H}(0)t}(\boldsymbol{y} - \boldsymbol{u}(0))$.

- *Let $\hat{f}_\infty$ be the limit of $\hat{f}_t$. Then $\hat{f}_\infty \in \mathcal{F}_B$ defined by*
$$\mathcal{F}_B := \left\{ f : \boldsymbol{x} \mapsto k_0(\boldsymbol{x}, \boldsymbol{X})\boldsymbol{\alpha} : \boldsymbol{\alpha}^T \boldsymbol{H}(0)\boldsymbol{\alpha} \leq B^2 \right\} \tag{21}$$
  *where $B^2 := (\boldsymbol{y} - \boldsymbol{u}(0))^T \boldsymbol{H}(0)^{-1}(\boldsymbol{y} - \boldsymbol{u}(0))$.*

- *The Rademacher complexity of $\mathcal{F}_B$ is given by*
$$\mathfrak{R}_S(\mathcal{F}_B) \leq c\sqrt{\frac{(\boldsymbol{y} - \boldsymbol{u}(0))^T \boldsymbol{H}(0)^{-1}(\boldsymbol{y} - \boldsymbol{u}(0))}{N}}. \tag{22}$$

It is clear that both $f_t$ and $\hat{f}_t$ are close related to the dynamics of training estimation $\boldsymbol{u}(t)$ and $\hat{\boldsymbol{u}}(t)$, correspondingly. By equation 15, the following lemma shows $\boldsymbol{u}(t)$ can be considered as a perturbed version of $\hat{\boldsymbol{u}}(t)$.

**Lemma 4.3** *Suppose the assumptions made in Theorem 3.1 holds. Then for all $t \geq 0$ we have*
$$\|\boldsymbol{u}(t) - \hat{\boldsymbol{u}}(t)\| = \mathcal{O}\left(\frac{n^{3/2}}{\lambda_0 \delta m^{1/4}} e^{-\lambda_0 t}\right), \tag{23}$$

The following lemma show $f_t$ is close to $\hat{f}_t$ if the implicit network is sufficiently overparameterized.

**Lemma 4.4** *Suppose the assumptions made in Theorem 3.1 holds. Then with probability at least $1 - \delta$ over random initialization equation 4, for all $t \geq 0$ and $\|\boldsymbol{x}_0\| = 1$ we have*
$$\left| f_t(\boldsymbol{x}_0) - \hat{f}_t(\boldsymbol{x}_0) \right| = \mathcal{O}\left(\frac{n^2}{\lambda_0^2 \delta m^{1/4}}\right). \tag{24}$$

Since Theorem 4.1 assumes $m = \Omega(\lambda_0^{-8}\delta^{-1}n^{10})$, Lemma 4.4 immediately implies
$$\left| f_t(\boldsymbol{x}_0) - \hat{f}_t(\boldsymbol{x}_0) \right| = \mathcal{O}\left(\sqrt{\frac{(\boldsymbol{y} - \boldsymbol{u}(0))^T \boldsymbol{H}(0)^{-1}(\boldsymbol{y} - \boldsymbol{u}(0))}{N}}\right)$$

Clearly, the RHS in equation 24 is independent of time $t$. Let $t \to \infty$ and we obtain the limiting functions $f_\infty$ and $\hat{f}_\infty$. Combining the Rademacher complexity in Lemma 4.2 with the bound in Lemma 4.4, we can complete the proof.

## 5 DISCUSSION ON INITIALIZATION

In this section, we want to show that the assumptions made in Theorem 3.1 and 4.1 can be satisfied by some specified random initialization with exponentially high probability. Unfortunately, these random initialization are not commonly used in practice. In addition, to our best knowledge, no previous works are studying the limiting neural tangent kernel $\boldsymbol{H}^\infty := \lim_{m \to \infty} \boldsymbol{H}(0)$ for implicit neural networks. Although this problem has been studied for some standard feed-forward neural networks, the case of implicit neural networks is more complicated since they have infinite depth. We will consider that as our future work.

By using simple linear algebra results for PSD matrices, the Gram matrix $\boldsymbol{H}(0)$ satisfies the following inequality

$$\begin{aligned}
\boldsymbol{H}(0) &\succeq \|\boldsymbol{b}(0)\|^2 \lambda_{\min}\left\{\boldsymbol{Q}(0)^{-1}\boldsymbol{Q}(0)^{-T}\right\} \cdot \frac{\gamma^2}{m}\boldsymbol{D}(0)\left[\boldsymbol{I} \otimes \boldsymbol{Z}(0)\boldsymbol{Z}(0)^T\right]\boldsymbol{D}(0) \\
&\succeq \frac{c}{m}\boldsymbol{D}(0)\left[\boldsymbol{I} \otimes \boldsymbol{Z}(0)\boldsymbol{Z}(0)^T\right]\boldsymbol{D}(0)
\end{aligned} \tag{25}$$

where $c > 0$ absorbs constants related to $\gamma_0$ and "$\boldsymbol{A} \succeq \boldsymbol{B}$" if $\boldsymbol{A} - \boldsymbol{B}$ is PSD. To show $\lambda_{\min}[\boldsymbol{H}(0)]$ is lower bounded, it suffices to show the RHS is strictly positive (with high probability). In the following, we provide a simple example of random initialization by which the RHS in equation 25 is strictly positive.

## 5.1 $\lambda_{\min}\{\boldsymbol{H}(0)\} \geq \lambda_0 > 0$

We choose $\phi = \sigma$ to also be the ReLU activation function. The random initialization equation 4 is specified as follows

$$\boldsymbol{a}_r \overset{i.i.d.}{\sim} |\mathcal{N}|\,(0, \boldsymbol{I}_m), \quad \boldsymbol{w}_r \overset{i.i.d.}{\sim} \mathcal{N}(\boldsymbol{0}, \boldsymbol{I}_d), \tag{26}$$

where $\mathcal{N}$ is Gaussian distribution and $|\mathcal{N}|$ is half-normal distribution. Then $\boldsymbol{A}_{ij}(0) \geq 0$ for all $i, j$. It follows from the nonnegativity of ReLU activation that $\boldsymbol{\Phi}_{ij} \geq 0$ for all $i, j$. By Neumann series, we can write the explicit form of the equilibrium point $\boldsymbol{Z}(0)$ at initialization as follows

$$\boldsymbol{Z}(0) = \sigma(\boldsymbol{X}\boldsymbol{W}(0))\left[\boldsymbol{I}_m - \gamma\boldsymbol{A}(0)\right]^{-1},$$

and $\boldsymbol{D}(0) = \boldsymbol{I}_{nm}$. The inequality equation 25 becomes

$$\boldsymbol{H}(0) \succeq \frac{c}{m}\sum_{r=1}^{m}\sigma(\boldsymbol{X}\boldsymbol{w}_r(0))\sigma(\boldsymbol{X}\boldsymbol{w}_r(0))^T. \tag{27}$$

Let $m \to \infty$, then we obtain the kernel matrix given by

$$\boldsymbol{G}^\infty := \mathbb{E}_{\boldsymbol{w}\sim\mathcal{N}(0,\boldsymbol{I}_d)}\left[\sigma(\boldsymbol{X}\boldsymbol{w})\sigma(\boldsymbol{X}\boldsymbol{w})^T\right] \tag{28}$$

that is induced by the neural tangent kernel Jacot et al. (2018) defined by

$$k_*(\boldsymbol{x}, \boldsymbol{x}') := \mathbb{E}_{\boldsymbol{w}\sim\mathcal{N}(0,\boldsymbol{I}_d)}\left[\sigma(\boldsymbol{w}^T\boldsymbol{x})\sigma(\boldsymbol{w}^T\boldsymbol{x}')\right].$$

Several previous works have shown that $\lambda_* := \lambda_{\min}\{\boldsymbol{G}^\infty\} > 0$ under some mild data distribution assumption. For example, Gao et al. (2021) shows that $\lambda_* > 0$ if no two training points are parallel. By using Matrix-Chernoff inequality, one can easily show

$$\lambda_{\min}\{\boldsymbol{H}(0)\} \geq \frac{c}{m}\lambda_{\min}\left[\sigma(\boldsymbol{X}\boldsymbol{W}(0))\sigma(\boldsymbol{X}\boldsymbol{W}(0))^T\right] \geq \frac{c}{m} \cdot m\lambda_*/4 = c\lambda_*/4 := \lambda_0.$$

with probability at least $1 - \delta$ (see Lemma 5.2 of Nguyen et al. (2021)) if $m = \tilde{\Omega}(n/\lambda_*)$ holds, where $\tilde{\Omega}$ omits logarithmic factors depending on $\delta$. Thus, the conditions on Theorem 3.1 and 4.1 are all satisfied with probability at least $1 - \delta$. By using union bound, the results in Theorem 3.1 and 4.1 are obtained.

## 5.2 Generalization bound induced by $\boldsymbol{H}^\infty$

To simplify the analysis, we assume in this subsection that $\boldsymbol{A}(0)$ is a randomly initialized diagonal matrix, e.g., $\boldsymbol{A}_{ij}(0) = \delta_{ij}z$ with $z \sim \mathcal{N}(0, 1)$. Then

$$\boldsymbol{H}(0)_{ij} = \frac{\gamma^2(\boldsymbol{z}_i^T\boldsymbol{z}_j)}{m}\sum_{r=1}^{m}\hat{b}_{ir}\hat{b}_{jr}\sigma'(\gamma\boldsymbol{a}_r^T\boldsymbol{z}_i + \phi(\boldsymbol{w}_r^T\boldsymbol{x}_i))\sigma'(\gamma\boldsymbol{a}_r^T\boldsymbol{z}_j + \phi(\boldsymbol{w}_r^T\boldsymbol{x}_j))$$

where $\hat{b}_{ir}$ is the $r$-the entry of the vector $\hat{\boldsymbol{b}}_i = \boldsymbol{Q}_i^{-1}\boldsymbol{b}$. Since $\boldsymbol{A}(0)$ is diagonal, we have $\hat{b}_{ir} = \Theta(1)$ for all $i \in [n], r \in [m]$. It follows from (Gao et al., 2021, Lemma 2.2) that $\|\boldsymbol{z}_i\| \leq \gamma^{-1}$ for all $i \in [n]$. Therefore, $\boldsymbol{H}(0)_{ij}$ is the average of $m$ random variables that are bounded in $[-c, c]$ for some constant $c > 0$. By Hoeffding's inequality, we have

$$\|\boldsymbol{H}(0) - \boldsymbol{H}^\infty\|_F = \mathcal{O}\left(\frac{n\sqrt{\log(n/\delta)}}{\sqrt{m}}\right).$$

Therefore, the generalization bound in Theorem 4.1 becomes

$$R(f_\infty) \leq \mathcal{O}\left(\sqrt{\frac{\boldsymbol{y}^T(\boldsymbol{H}^\infty)^{-1}\boldsymbol{y}}{n}} + \sqrt{\frac{\log(n/\delta)}{n}}\right).$$

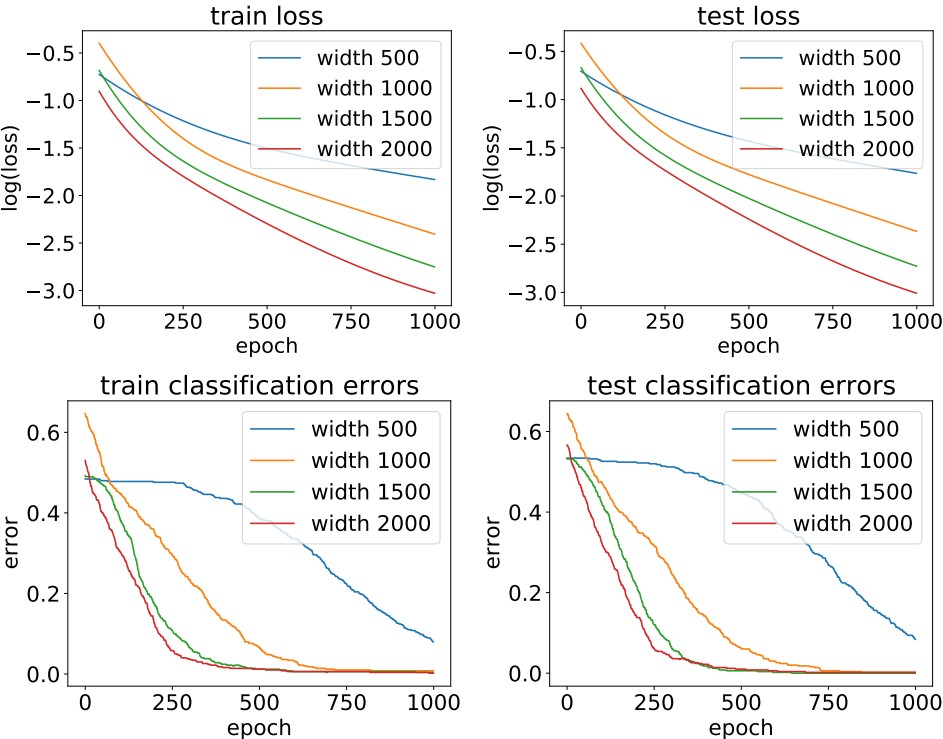

Figure 1: We evaluate the impact of the **width** $m$ on the training loss, test loss, and operator norm of the scaled matrix $\gamma A(k)$ on the modified dataset of MNIST.

This is one of the well-known generalization bounds for finite-depth neural networks Arora et al. (2019); Cao & Gu (2020). Since $\boldsymbol{H}^\infty$ is constant and induced from the ReLu activation, one of the advantages of this generalization bound is that it can be computed directly by using the training sample $S = \{(\boldsymbol{x}_i, y_i)\}_{i=1}^n$. In contrast, the generalization bound in Theorem 4.1 is initialization-dependent, which probably has the potential to provide an estimate of the test performance closely related to empirical results.

## 6 EXPERIMENTAL RESULTS

We conduct a series of experiments to verify the convergence and test the performance of gradient flow. The experimental setup is identical to Gao et al. (2021), but we only train the implicit layer: we draw 500 random samples of classes 0 and 1 from the MNIST dataset. We can see from Figure 1 that the training loss (*i.e.*, square loss) and train classification error (*i.e.*, 0-1 loss) is consistently decreased to 0. In addition, a faster convergence rate is obtained by using wider implicit layers. As we prove in Theorem 4.1, overparameterized implicit neural network also generalizes. This can be seen from Figure 1 as the test loss and test classification errors are reduced consistently. It is worth noting that a wider implicit network also provides better test performance.

## 7 CONCLUSION

This paper establishes a convergence result of gradient flow for implicit neural networks. We show that gradient flow with random initialization converges to a global minimum at a linear rate, even if we only optimize the implicit layer while the rest is untrained. Moreover, we prove that overparameterized implicit network is closely related to a kernel machine. By leveraging the Rademacher complexity theory, an initialization-sensitive generalization bound is provided. Thus, an arbitrarily small generalization error can be obtained if the implicit layer is sufficiently overparameterized.

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

# A    USEFUL MATHEMATICAL LEMMAS

**Lemma A.1 (Gronwall's inequality)** *Let $u$, $\alpha$, $\beta$ be real-valued continuous functions that satisfies the integral inequality*

$$u(t) \leq \alpha(t) + \int_0^t \beta(s)u(s)ds. \tag{29}$$

*Then*

$$u(t) \leq \alpha(t) + \int_0^t \alpha(s)\beta(s) \exp\left(\int_s^t \beta(r)dr\right) ds. \tag{30}$$

*If, in addition, $\alpha(t)$ is non-decreasing, then*

$$u(t) \leq \alpha(t) \exp\left(\int_0^t \beta(s)ds\right) \tag{31}$$

**Lemma A.2 (Weyl's inequality)** *Let $\boldsymbol{A}, \boldsymbol{B} \in \mathbb{R}^{m \times n}$ with singular values $\sigma_1(\boldsymbol{A}) \geq \cdots \geq \sigma_r(\boldsymbol{A})$, where $r := \min\{m, n\}$. Then*

$$|\sigma_k(\boldsymbol{A}) - \sigma_k(\boldsymbol{A})| \leq \|\boldsymbol{A} - \boldsymbol{B}\| \tag{32}$$

**Theorem A.3** *Let $k$ be a kernel and $S = \{\boldsymbol{X}, \boldsymbol{y}\}$ be a sample of size $n$ for which $k(\boldsymbol{x}, \boldsymbol{x}') \leq r^2$. Let $\mathcal{F}_B = \{\boldsymbol{x} \mapsto k(\boldsymbol{x}, \boldsymbol{X})\boldsymbol{\alpha} : \boldsymbol{\alpha}^T k(\boldsymbol{X}, \boldsymbol{X})\boldsymbol{\alpha} \leq B^2\}$. Then*

$$\mathfrak{R}_S(\mathcal{F}_B) \leq \frac{B}{n}\sqrt{\mathbf{tr}(k(\boldsymbol{X}, \boldsymbol{X}))} \leq \frac{Br}{\sqrt{n}} \tag{33}$$

**Theorem A.4** *Suppose the loss function $\ell(y, \cdot)$ is $\rho$-Lipschitz continuous and is bounded in $[0, c]$ for some constant $c > 0$. Then with probability at least $1 - \delta$ over random sample $S$ of size $N$*

$$\sup_{f \in \mathcal{F}} [R(f) - R_S(f)] \leq 2\mathfrak{R}_S(\mathcal{F}) + 3c\sqrt{\frac{\log(2/\delta)}{2N}}.$$

**Lemma A.5** *Suppose $\|\boldsymbol{A}\| \leq M$ for some constant $M > 0$ and choose the scalar $\gamma > 0$ such that $\gamma_0 := \gamma M < 1$. Then the existence of the fixed point $\boldsymbol{Z}$ is uniquely determined. Moreover, we have $\|\boldsymbol{Z}^\ell\|_F \leq \frac{1}{1-\gamma_0}\|\Phi\|_F$ for all $\ell$, hence $\|\boldsymbol{Z}\|_F \leq \frac{1}{1-\gamma_0}\|\Phi\|_F$.*

# B    PROOF OF LEMMA 3.2

It follows from Lemma A.5 that $\boldsymbol{Z}$ is uniquely determined. As a result, $\boldsymbol{Z}$ is also a root of the following function

$$f(\boldsymbol{Z}, \boldsymbol{A}) = \boldsymbol{Z} - \sigma(\gamma \boldsymbol{Z} \boldsymbol{A} + \boldsymbol{\Phi}).$$

Then by using the implicit function theorem, (Gao & Gao, 2022, Lemma 3.2) provides the derivative of $\boldsymbol{Z}$ with respect to $\boldsymbol{A}$ as follows

Then the differential of $f$ is given by

$$\begin{aligned} df &= d\boldsymbol{Z} - d\sigma(\gamma \boldsymbol{Z} \boldsymbol{A} + \boldsymbol{\Phi}) \\ &= d\boldsymbol{Z} - \sigma'(\boldsymbol{U}) \odot d(\gamma \boldsymbol{Z} \boldsymbol{A} + \boldsymbol{\Phi}) \\ &= d\boldsymbol{Z} - \sigma'(\boldsymbol{U}) \odot \gamma(d\boldsymbol{Z})\boldsymbol{A} - \sigma'(\boldsymbol{U}) \odot \gamma \boldsymbol{Z} d\boldsymbol{A} \end{aligned}$$

where $\boldsymbol{U} := \gamma \boldsymbol{Z} \boldsymbol{A} + \boldsymbol{\Phi}$. Taking vectorization on both sides yields

$$\text{vec}\,(df) = \text{vec}\,(d\boldsymbol{Z}) - \gamma \boldsymbol{D}\left(\boldsymbol{A}^T \otimes \boldsymbol{I}_n\right) \text{vec}\,(d\boldsymbol{Z}) - \gamma \boldsymbol{D}\left(\boldsymbol{I}_m \otimes \boldsymbol{Z}\right) \text{vec}\,(d\boldsymbol{A})$$

where $D := \mathbf{diag}[\text{vec}\,(\sigma'(U))]$. Thus, we obtain

$$\begin{aligned} \frac{\partial f}{\partial \boldsymbol{Z}} &= \left[\boldsymbol{I}_{Nm} - \gamma \boldsymbol{D}(\boldsymbol{A}^T \otimes \boldsymbol{I}_n)\right]^T \\ \frac{\partial f}{\partial \boldsymbol{A}} &= -\gamma \left[\boldsymbol{D}(\boldsymbol{I}_m \otimes \boldsymbol{Z})\right]^T \end{aligned}$$

By reverse triangle inequality, we have

$$\left\| \boldsymbol{I}_{nm} - \gamma \boldsymbol{D}(\boldsymbol{A}^T \otimes \boldsymbol{I}_n) \right\| \geq 1 - \gamma \| \boldsymbol{D}(\boldsymbol{A}^T \otimes \boldsymbol{I}_n) \| \geq 1 - \gamma \| \boldsymbol{A} \| > 1 - \gamma_0 > 0, \qquad (34)$$

where we use $\| \boldsymbol{A} \| \leq M$ and $\gamma_0 = \gamma / M < 1$. Hence, the matrix $\boldsymbol{Q} := \left[ \boldsymbol{I}_{nm} - \gamma \boldsymbol{D}(\boldsymbol{A}^T \otimes \boldsymbol{I}_n) \right]$ is invertible.

Since the fixed point $\boldsymbol{Z}$ is the root of $f$ and the matrix $\boldsymbol{Q}$ is invertible, by using the *implicit function theorem*, we have

$$\frac{\partial \boldsymbol{Z}}{\partial \boldsymbol{A}} \frac{\partial f}{\partial \boldsymbol{Z}} + \frac{\partial f}{\partial \boldsymbol{A}} = 0,$$

which implies

$$\frac{\partial \boldsymbol{Z}}{\partial \boldsymbol{A}} = - \left( \frac{\partial f}{\partial \boldsymbol{A}} \right) \left( \frac{\partial f}{\partial \boldsymbol{Z}} \right)^{-1} = \gamma \left[ \boldsymbol{D}(\boldsymbol{I}_m \otimes \boldsymbol{Z}) \right]^T \boldsymbol{Q}^{-T}.$$

Next, we derive the derivative of $\boldsymbol{J}^T$ with respect to $\boldsymbol{A}$. The differential of $\boldsymbol{J}^T$ is given by

$$\begin{aligned} d\boldsymbol{J}^T =& d\boldsymbol{D}\boldsymbol{U}^{-1}\boldsymbol{B} \\ =& (d\boldsymbol{D})\boldsymbol{U}^{-1}\boldsymbol{B} - \boldsymbol{D}\boldsymbol{U}^{-1}(d\boldsymbol{U})\boldsymbol{U}^{-1}\boldsymbol{B} \\ =& \gamma \boldsymbol{D}\boldsymbol{U}^{-1}(d\boldsymbol{A})\boldsymbol{D}\boldsymbol{U}^{-1}\boldsymbol{B} \end{aligned}$$

where we use the fact $\sigma''(z) = 0$ and we denote $\boldsymbol{U} := \boldsymbol{Q}^T$ and $\boldsymbol{B} := \boldsymbol{b} \otimes \boldsymbol{I}_n$ to simplify derivation. Take vectorization on both sides and we can obtain the derivatives of $\boldsymbol{J}^T$ with respective to $\boldsymbol{A}$ as follows

$$\frac{\partial \boldsymbol{J}^T}{\partial \boldsymbol{A}} = \left[ (\boldsymbol{D}\boldsymbol{U}^{-1}\boldsymbol{B})^T \otimes \gamma \boldsymbol{D}\boldsymbol{U}^{-1} \right]^T$$

## C  Proof of Theorem 3.1

Note that at the initialization, we have

$$\begin{aligned} \mathbb{E} \left| f_\theta(\boldsymbol{x}) \right|^2 =& \mathbb{E}(\boldsymbol{z}^T \boldsymbol{b})^2 = \mathbb{E}\left[ \sum_{r=0}^m \sum_{s=0}^m z_r z_s b_r b_s \right] \\ =& \frac{1}{m} \mathbb{E}\left[ \sum_{r=0}^m z_r^2 \right], \quad b_r \sim \mathcal{U}\{-1/\sqrt{m}, +1\sqrt{m}\} \\ =& \frac{1}{m} \mathbb{E} \| \boldsymbol{z} \|^2 \\ \leq& \frac{1}{m} \frac{1}{(1-\gamma_0)^2} \mathbb{E} \| \boldsymbol{\phi} \|^2, \quad \text{Lemma A.5} \\ \leq& \frac{1}{m} \frac{1}{(1-\gamma_0)^2} \mathbb{E} \| \boldsymbol{W}^T \boldsymbol{x} \|^2, \quad \phi(\cdot) \text{ is 1-Lipschitz continuous} \\ \leq& \frac{1}{m} \frac{1}{(1-\gamma_0)^2} \boldsymbol{x}^T \left[ \sum_{r=0}^m \mathbb{E}(\boldsymbol{w}_r \boldsymbol{w}_r^T) \right] \boldsymbol{x} \\ =& \frac{1}{(1-\gamma_0)^2} \end{aligned}$$

By using Markov's inequality, we have

$$\mathbb{P}\left[ \| \boldsymbol{u}(0) \|^2 \geq \epsilon \right] \leq \epsilon^{-1} \mathbb{E} \| \boldsymbol{u}(0) \|^2 \leq \epsilon^{-1} N (1-\gamma_0)^{-2} = \delta.$$

Thus, we obtain $\| \boldsymbol{u}(0) \| = \mathcal{O}\left( \sqrt{N/\delta} \right)$ with probability at least $1 - \delta$.

Now we are ready to present the convergence proof. By (Vershynin, 2018, Theorem 4.4.5), we know $\| \boldsymbol{A}(0) \| \leq c\sqrt{m}$ and $\| \boldsymbol{W}(0) \| \leq \sqrt{m}$ for some constant $c > 0$ with probability at least $1 - \delta$. Without loss generalization, we assume $c = 1$. Since we choose $\gamma := \gamma_0/\sqrt{m}$, the equilibrium point $\boldsymbol{Z}(0)$ is well defined. In the following, we can show $\| \boldsymbol{A}(t) \| = \mathcal{O}\left( \sqrt{m} \right)$ for all $t \geq 0$ and so $\boldsymbol{Z}(t)$ is well defined throughout training. Assume the results hold for all $0 \leq s \leq t$:

- $\boldsymbol{A}(s) = \mathcal{O}\left(\sqrt{m}\right)$
- $\|\boldsymbol{u}(s) - \boldsymbol{y}\|^2 \le e^{-\lambda_0 t}\|\boldsymbol{u}(0) - \boldsymbol{y}\|^2$

We can bound $\partial L / \partial \boldsymbol{A}$ as follows

$$
\begin{aligned}
\|\partial L(s)/\partial \boldsymbol{A}\| &= \left\| \frac{\gamma}{\sqrt{m}} \left[ \boldsymbol{D}(s)(\boldsymbol{I}_m \otimes \boldsymbol{Z}(s)) \right]^T \boldsymbol{Q}(s)^{-T} \left( \boldsymbol{b}^T \otimes \boldsymbol{I}_n \right)^T (\boldsymbol{u} - \boldsymbol{y}) \right\| \\
&\le \frac{c}{m} \|\boldsymbol{Z}(s)\|_F \|\boldsymbol{b}\| \|\boldsymbol{u}(s) - \boldsymbol{y}\| \\
&\le \frac{c}{m} \|\boldsymbol{W}\| \|\boldsymbol{X}\|_F \|\boldsymbol{b}\| \|\boldsymbol{u}(s) - \boldsymbol{y}\|, \quad \text{By Lemma A.5} \\
&\le c \|\boldsymbol{X}\|_F \|\boldsymbol{u}(s) - \boldsymbol{y}\|.
\end{aligned}
$$

Since $\|\boldsymbol{W}\| \le \sqrt{m}$ and $\|\boldsymbol{b}\| = \sqrt{m}$, we have

$$
\|\partial L(s)/\partial \boldsymbol{A}\| \le c\|\boldsymbol{X}\|_F \|\boldsymbol{u}(s) - \boldsymbol{y}\|. \tag{35}
$$

Furthermore, we have

$$
\|\boldsymbol{A}(t) - \boldsymbol{A}(0)\|_F \le \int_0^t \|\partial L/\partial A(s)\| \, ds \le \frac{c}{\lambda_0} \|\boldsymbol{X}\|_F \|\boldsymbol{u}(0) - \boldsymbol{y}\| \le \sqrt{m}
$$

where the last inequality is because $m = \Omega\left( \frac{N^2}{\lambda_0^2 \delta} \right)$. As a result, we have

$$
\|\boldsymbol{A}(t)\| \le \|\boldsymbol{A}(t) - \boldsymbol{A}(0)\|_F + \|\boldsymbol{A}(0)\| = \mathcal{O}\left(\sqrt{m}\right).
$$

Note that

$$
\begin{aligned}
&\|\boldsymbol{H}(t) - \boldsymbol{H}(0)\| \\
={}& \frac{\gamma^2}{m} \|\boldsymbol{J}(t)(\boldsymbol{I}_m \otimes \boldsymbol{Z}(t)\boldsymbol{Z}(t)^T)\boldsymbol{J}(t)^T - \boldsymbol{J}(0)(\boldsymbol{I}_m \otimes \boldsymbol{Z}(0)\boldsymbol{Z}(0)^T)\boldsymbol{J}(0)^T\| \\
\le{}& \frac{\gamma^2}{m} \|\boldsymbol{Z}(t)\|_F^2 \|\boldsymbol{J}(t) - \boldsymbol{J}(0)\| \|\boldsymbol{J}(t)\| + \frac{\gamma^2}{m} \|\boldsymbol{Z}(t)\boldsymbol{Z}(t)^T - \boldsymbol{Z}(0)\boldsymbol{Z}(0)^T\| \|\boldsymbol{J}(0)\| \|\boldsymbol{J}(t)\| \\
&+ \frac{\gamma^2}{m} \|\boldsymbol{Z}(0)\|_F^2 \|\boldsymbol{J}(t) - \boldsymbol{J}(0)\| \|\boldsymbol{J}(0)\|
\end{aligned}
$$

In the following, we will bound each terms.

By Lemma 3.2, we can bound $d\mathrm{vec}\left(\boldsymbol{Z}\right)/dt$ as follows

$$
\left\| \frac{d\mathrm{vec}\left(\boldsymbol{Z}\right)}{dt} \right\| = \left\| \left( \frac{\partial \boldsymbol{Z}}{\partial \boldsymbol{A}} \right)^T \left( -\frac{\partial L}{\partial \boldsymbol{A}} \right) \right\| = \mathcal{O}\left( \|\boldsymbol{X}\|_F^2 \|\boldsymbol{u}(t) - \boldsymbol{y}\| \right)
$$

Thus, we can further bound $\boldsymbol{Z}(t) - \boldsymbol{Z}(0)$ as follows

$$
\|\boldsymbol{Z}(t) - \boldsymbol{Z}(0)\|_F \le \int_0^t \left\| \frac{d\mathrm{vec}\left(\boldsymbol{Z}\right)}{ds} \right\| ds \le \frac{c}{\lambda_0} \|\boldsymbol{X}\|_F^2 \|\boldsymbol{u}(0) - \boldsymbol{y}\|.
$$

Similarly, we can bound the dynamics of $\boldsymbol{J}^T$ as follows

$$
\frac{d\mathrm{vec}\left(\boldsymbol{J}^T\right)}{dt} \le \left\| \left[ \frac{\partial \boldsymbol{J}^T}{\partial \boldsymbol{A}} \right]^T \right\| \left\| \left( \frac{\partial L}{\partial \boldsymbol{A}} \right) \right\| \le c\|\boldsymbol{X}\|_F \|\boldsymbol{u}(t) - \boldsymbol{y}\|.
$$

Therefore, we can bound $\boldsymbol{J}(t) - \boldsymbol{J}(0)$ as follows

$$
\|\boldsymbol{J}(t) - \boldsymbol{J}(0)\| \le \int_0^t \left\| \frac{d\mathrm{vec}\left(\boldsymbol{J}^T\right)}{ds} \right\| ds \le \frac{c}{\lambda_0} \|\boldsymbol{X}\|_F \|\boldsymbol{u}(0) - \boldsymbol{y}\|,
$$

Combining equation 12, equation 13, equation 14, and Lemma A.5, we have

$$
\|\boldsymbol{H}(t) - \boldsymbol{H}(0)\| \le \frac{c\|\boldsymbol{X}\|_F^3 \|\boldsymbol{u}(0) - \boldsymbol{y}\|}{\lambda_0 \sqrt{m}} \le \frac{cN^2}{\lambda_0 \sqrt{\delta}\sqrt{m}} \le \lambda_0/2,
$$

where the last inequality follows from $m = \Omega\left(\frac{N^4}{\lambda_0^4 \delta}\right)$. By Weyl's inequality, we have

$$\lambda_{\min}\left[\boldsymbol{H}(t)\right] \geq \lambda_{\min}\left[\boldsymbol{H}(0)\right] - \|\boldsymbol{H}(t) - \boldsymbol{H}(0)\| \geq \lambda_0/2.$$

Therefore, we have

$$
\begin{aligned}
\frac{d}{dt}\|\boldsymbol{u} - \boldsymbol{y}\|^2 &= 2(\boldsymbol{u} - \boldsymbol{y})^T \frac{d\boldsymbol{u}}{dt} \\
&= -2(\boldsymbol{u} - \boldsymbol{y})^T \boldsymbol{H}(t)(\boldsymbol{u} - \boldsymbol{y}) \\
&\geq -\lambda_0 \|\boldsymbol{u} - \boldsymbol{y}\|^2.
\end{aligned}
$$

By solving the ordinary differential equation above, we have

$$\|\boldsymbol{u}(t) - \boldsymbol{y}\|^2 \leq e^{-\lambda_0 t}\|\boldsymbol{u}(0) - \boldsymbol{y}\|^2.$$

## D    PROOF OF LEMMA 4.2

The dynamics of $\boldsymbol{y} - \hat{\boldsymbol{u}}(t)$ is given by

$$\frac{d\left(\boldsymbol{y} - \hat{\boldsymbol{u}}\right)}{dt} = -\boldsymbol{H}(0)(\boldsymbol{y} - \hat{\boldsymbol{u}}(t))$$

Solving the ordinary differential equation above yields

$$\boldsymbol{y} - \hat{\boldsymbol{u}}(t) = e^{\int_0^t -\boldsymbol{H}(0)ds}(\boldsymbol{y} - \hat{\boldsymbol{u}}(0)) = e^{-\boldsymbol{H}(0)t}(\boldsymbol{y} - \hat{\boldsymbol{u}}(0)).$$

Let $\boldsymbol{H}(0) = \boldsymbol{Q}\boldsymbol{\Lambda}\boldsymbol{Q}^T$ be the eigenvalue decomposition of $\boldsymbol{H}(0)$, then we have

$$
\begin{aligned}
\hat{f}_t(\boldsymbol{x}_0) &= f_0(\boldsymbol{x}_0) + \int_0^t k_0(\boldsymbol{x}_0, \boldsymbol{X})(\boldsymbol{y} - \hat{\boldsymbol{u}}(s))ds \\
&= f_0(\boldsymbol{x}_0) + k_0(\boldsymbol{x}_0, \boldsymbol{X}) \int_0^t e^{-\boldsymbol{H}(0)s}(\boldsymbol{y} - \boldsymbol{u}(0))ds \\
&= f_0(\boldsymbol{x}_0) + k_0(\boldsymbol{x}_0, \boldsymbol{X})\boldsymbol{Q}\left(\int_0^t e^{-\boldsymbol{\Lambda}s}ds\right)\boldsymbol{Q}^T(\boldsymbol{y} - \boldsymbol{u}(0))
\end{aligned}
$$

Let $t \to \infty$, then we have

$$\hat{f}_\infty(\boldsymbol{x}_0) = f_0(\boldsymbol{x}_0) + k_0(\boldsymbol{x}_0, \boldsymbol{X})\boldsymbol{H}(0)^{-1}(\boldsymbol{y} - \boldsymbol{u}(0)).$$

Let $\boldsymbol{\alpha} := \boldsymbol{H}(0)^{-1}(\boldsymbol{y} - \boldsymbol{u}(0))$, then we have

$$B^2 := \boldsymbol{\alpha}^T \boldsymbol{H}(0)\boldsymbol{\alpha} = (\boldsymbol{y} - \boldsymbol{u}(0))^T \boldsymbol{H}(0)^{-1}(\boldsymbol{y} - \boldsymbol{u}(0)).$$

Note that

$$k_0(\boldsymbol{x}, \boldsymbol{x}) = \left\langle \frac{\partial f_0(\boldsymbol{x})}{\partial \boldsymbol{A}}, \frac{\partial f_0(\boldsymbol{x})}{\partial \boldsymbol{A}} \right\rangle \leq \left\| \frac{\partial f_0(\boldsymbol{x})}{\partial \boldsymbol{A}} \right\|^2 = \mathcal{O}\left(\|\boldsymbol{x}\|^2\right). \tag{36}$$

Therefore, the Rademacher complexity of $\mathcal{F}_B$ is given by

$$\Re_S(\mathcal{F}_B) \leq \frac{B}{n}\sqrt{\mathbf{tr}(\boldsymbol{H}(0))} \leq c\sqrt{\frac{(\boldsymbol{y} - \boldsymbol{u}(0))^T \boldsymbol{H}(0)^{-1}(\boldsymbol{y} - \boldsymbol{u}(0))}{n}}.$$

## E    PROOF OF LEMMA 4.3

The dynamics of $\boldsymbol{u}(t) - \hat{\boldsymbol{u}}(t)$ is given by

$$
\begin{aligned}
\frac{d(\boldsymbol{u} - \hat{\boldsymbol{u}})}{dt} &= \boldsymbol{H}(t)(\boldsymbol{y} - \boldsymbol{u}) - \boldsymbol{H}(0)(\boldsymbol{y} - \hat{\boldsymbol{u}}) \\
&= [\boldsymbol{H}(t) - \boldsymbol{H}(0)](\boldsymbol{y} - \boldsymbol{u}) - \boldsymbol{H}(0)(\boldsymbol{u} - \hat{\boldsymbol{u}})
\end{aligned}
$$

Then we have

$$
\begin{aligned}
\frac{d}{dt}\|\boldsymbol{u} - \hat{\boldsymbol{u}}\|^2 &= 2(\boldsymbol{u} - \hat{\boldsymbol{u}})\frac{d(\boldsymbol{u} - \hat{\boldsymbol{u}})}{dt} \\
&= 2(\boldsymbol{u} - \hat{\boldsymbol{u}})^T\left[\boldsymbol{H}(t) - \boldsymbol{H}(0)\right](\boldsymbol{y} - \boldsymbol{u}) - 2(\boldsymbol{u} - \hat{\boldsymbol{u}})^T\boldsymbol{H}(0)(\boldsymbol{u} - \hat{\boldsymbol{u}}) \\
&\leq 2\|\boldsymbol{u} - \hat{\boldsymbol{u}}\|\|\boldsymbol{H}(t) - \boldsymbol{H}(0)\|\|\boldsymbol{y} - \boldsymbol{u}\| - 2\lambda_0\|\boldsymbol{u} - \hat{\boldsymbol{u}}\|^2
\end{aligned}
$$

where we use $\lambda_{\min}[\boldsymbol{H}(0)] \geq \lambda_0 > 0$. By using Gronwall's inequality, we have

$$
\begin{aligned}
\|\boldsymbol{u}(t) - \hat{\boldsymbol{u}}(t)\|^2 &\leq \int_0^t 2\|\boldsymbol{u}(s) - \hat{\boldsymbol{u}}(s)\|\|\boldsymbol{H}(s) - \boldsymbol{H}(0)\|\|\boldsymbol{y} - \boldsymbol{u}(s)\|ds + \int_0^t (-2\lambda_0)\|\boldsymbol{u}(s) - \hat{\boldsymbol{u}}(s)\|^2 ds \\
&\leq \int_0^t 2\|\boldsymbol{u}(s) - \hat{\boldsymbol{u}}(s)\|\|\boldsymbol{H}(s) - \boldsymbol{H}(0)\|\|\boldsymbol{y} - \boldsymbol{u}(s)\|ds \cdot \exp\left(\int_0^t -(2\lambda_0)ds\right) \\
&= \int_0^t 2\|\boldsymbol{u}(s) - \hat{\boldsymbol{u}}(s)\|\|\boldsymbol{H}(s) - \boldsymbol{H}(0)\|\|\boldsymbol{y} - \boldsymbol{u}(s)\|ds \cdot e^{-2\lambda_0 t} \\
&\leq \frac{cN^2}{\lambda_0\sqrt{\delta}\sqrt{m}}\int_0^t \|\boldsymbol{u}(s) - \hat{\boldsymbol{u}}(s)\|\|\boldsymbol{y} - \boldsymbol{u}(s)\|ds \cdot e^{-2\lambda_0 t} \\
&\leq \frac{cN^2}{\lambda_0\sqrt{\delta}\sqrt{m}}\int_0^t \|\boldsymbol{y} - \hat{\boldsymbol{u}}(s)\|\|\boldsymbol{y} - \boldsymbol{u}(s)\|^2 ds \cdot e^{-2\lambda_0 t} \\
&\leq \frac{cN^2}{\lambda_0\sqrt{\delta}\sqrt{m}}\left[\int_0^t e^{-2\lambda_0 s}ds\right]\|\boldsymbol{y} - \boldsymbol{u}(0)\|^2 \cdot e^{-2\lambda_0 t} \\
&\leq \frac{cN^3}{\lambda_0^2\delta^2\sqrt{m}} \cdot e^{-2\lambda_0 t}
\end{aligned}
$$

Thus, we have

$$
\|\boldsymbol{u}(t) - \hat{\boldsymbol{u}}(t)\| = \mathcal{O}\left(\frac{N^{3/2}}{\lambda_0\delta m^{1/4}} \cdot e^{-\lambda_0 t}\right).
$$

## F   PROOF OF LEMMA 4.4

Note that

$$
\begin{aligned}
\frac{d(f_t(\boldsymbol{x}_0) - \hat{f}_t(\boldsymbol{x}_0))}{dt} &= k_t(\boldsymbol{x}_0, \boldsymbol{X})(\boldsymbol{y} - \boldsymbol{u}(t)) - k_0(\boldsymbol{x}_0, \boldsymbol{X})(\boldsymbol{y} - \hat{\boldsymbol{u}}(t)) \\
&= (k_t(\boldsymbol{x}_0, \boldsymbol{X}) - k_0(\boldsymbol{x}_0, \boldsymbol{X}))(\boldsymbol{y} - \boldsymbol{u}(t)) + k_0(\boldsymbol{x}_0, \boldsymbol{X})(\hat{\boldsymbol{u}}(t) - \boldsymbol{u}(t))
\end{aligned}
$$

In the following, we will bound each terms. Note that

$$
\begin{aligned}
&k_t(\boldsymbol{x}_i, \boldsymbol{x}_j) - k_0(\boldsymbol{x}_i, \boldsymbol{x}_j) \\
&= \frac{\gamma^2}{m}\left[(\boldsymbol{J}_i(t)\boldsymbol{J}_j(t)^T)(\boldsymbol{z}_i(t)^T\boldsymbol{z}_j(t)) - (\boldsymbol{J}_i()\boldsymbol{J}_j()^T)(\boldsymbol{z}_i(0)^T\boldsymbol{z}_j(0))\right] \\
&= \frac{\gamma^2}{m}\left[\boldsymbol{J}_i(t) - \boldsymbol{J}_i(0)\right]\boldsymbol{J}_j(t)^T(\boldsymbol{z}_i(t)^T\boldsymbol{z}_j(t)) + \frac{\gamma^2}{m}\boldsymbol{J}_i(0)\left[\boldsymbol{J}_j(t) - \boldsymbol{J}_j(0)\right]^T(\boldsymbol{z}_i(t)^T\boldsymbol{z}_j(t)) \\
&\quad + \frac{\gamma^2}{m}\boldsymbol{J}_i(0)\boldsymbol{J}_j(0)^T(\boldsymbol{z}_i(t) - \boldsymbol{z}_i(0))^T\boldsymbol{z}_j(t) + \frac{\gamma^2}{m}\boldsymbol{J}_i(0)\boldsymbol{J}_j(0)^T\boldsymbol{z}_i(0)^T[\boldsymbol{z}_j(t) - \boldsymbol{z}_j(0)]
\end{aligned}
$$

where

$$
\begin{aligned}
\boldsymbol{J}_i(t) &:= \boldsymbol{b}^T\boldsymbol{Q}_i(t)^{-1}\boldsymbol{D}_i(t), \\
\boldsymbol{Q}_i(t) &:= \boldsymbol{I}_m - \gamma\boldsymbol{D}_i(t)\boldsymbol{A}(t)^T, \\
\boldsymbol{D}_i(t) &:= \mathbf{diag}\left[\sigma'\left(\gamma\boldsymbol{A}^T\boldsymbol{z}_i(t) + \boldsymbol{\phi}_i\right)\right].
\end{aligned}
$$

By equation 13, equation 14, equation 12, and Lemma A.5, we have

$$
|k_t(\boldsymbol{x}_i, \boldsymbol{x}_j) - k_0(\boldsymbol{x}_i, \boldsymbol{x}_j)| = \mathcal{O}\left(\frac{\sqrt{N}}{\lambda_0\sqrt{\delta}\sqrt{m}}\right) \tag{37}
$$

where we also use the data assumption $\|\boldsymbol{x}_i\| = \|\boldsymbol{x}_j\| = 1$ and $\|\boldsymbol{u}(0) - \boldsymbol{y}\| = \mathcal{O}\left(\sqrt{N/\delta}\right)$. Thus, we obtain

$$\|k_t(\boldsymbol{x}_0, \boldsymbol{X}) - k_0(\boldsymbol{x}_0, \boldsymbol{X})\| = \mathcal{O}\left(\frac{N}{\lambda_0\sqrt{\delta}\sqrt{m}}\right). \tag{38}$$

Combining equation 38, equation 36, and equation 23 together, we have

$$\left|\frac{d(f_t(\boldsymbol{x}_0) - \hat{f}_t(\boldsymbol{x}_0))}{dt}\right| = \mathcal{O}\left(\frac{N^{3/2}}{\lambda_0\delta\sqrt{m}}e^{-\lambda_0 t}\right) + \mathcal{O}\left(\frac{N^2}{\lambda_0\delta m^{1/4}}e^{-\lambda_0 t}\right) = \mathcal{O}\left(\frac{N^2}{\lambda_0\delta m^{1/4}}e^{-\lambda_0 t}\right)$$

Therefore, we have

$$\left|f_t(\boldsymbol{x}_0) - \hat{f}_t(\boldsymbol{x}_0)\right| \leq \int_0^t \left|\frac{d(f_s(\boldsymbol{x}_0) - \hat{f}_s(\boldsymbol{x}_0))}{ds}\right| ds = \mathcal{O}\left(\frac{n^2}{\lambda_0^2\delta m^{1/4}}\right) \tag{39}$$

Since $m = \Omega(\lambda_0^{-8}\delta^{-1}N^{10})$, we have

$$\left|f_t(\boldsymbol{x}_0) - \hat{f}_t(\boldsymbol{x}_0)\right| = \mathcal{O}\left(\sqrt{\frac{(\boldsymbol{y} - \boldsymbol{u}(0))^T\boldsymbol{H}(0)^{-1}(\boldsymbol{y} - \boldsymbol{u}(0))}{N}}\right)$$

## G    PROOF OF THEOREM 4.1

Denote $C := \sqrt{(\boldsymbol{y} - \boldsymbol{u}(0))^T\boldsymbol{H}(0)^{-1}(\boldsymbol{y} - \boldsymbol{u}(0))/N}$. Let $\ell$ be a 1-Lipschitz-continuous loss function, we have

$$\left|\ell(y_0, f_\infty(\boldsymbol{x}_0)) - \ell(y_0, \tilde{f}_\infty(\boldsymbol{x}_0))\right| \leq \left|f_\infty(\boldsymbol{x}_0) - \tilde{f}_\infty(\boldsymbol{x}_0)\right| \leq C.$$

This implies that

$$\ell(y_0, f_\infty(\boldsymbol{x}_0)) \leq \ell(y_0, \tilde{f}_\infty(\boldsymbol{x}_0)) + C. \tag{40}$$

Since $(\boldsymbol{x}_0, y_0)$ is an arbitrary data point, taking expectation the inequality still holds:

$$R(f_\infty) \leq R(\tilde{f}_\infty) + C.$$

As a consequence, it follows from the Rademacher complexity theorem A.4 that we have

$$\begin{aligned}
R(f_\infty) \leq & R(\tilde{f}_\infty) + C \\
\leq & \hat{R}_S(\tilde{f}_\infty) + \mathfrak{R}_S(\mathcal{F}_B) + \sqrt{\frac{\log(1/\delta)}{N}} + C \\
= & \mathfrak{R}_S(\mathcal{F}_B) + \sqrt{\frac{\log(1/\delta)}{N}} + C, \quad \text{(i)} \\
\leq & \sqrt{\frac{(\boldsymbol{y} - \boldsymbol{u}(0))^T\boldsymbol{H}(0)^{-1}(\boldsymbol{y} - \boldsymbol{u}(0))}{N}} + \sqrt{\frac{\log(1/\delta)}{N}} + C \\
\leq & \mathcal{O}\left(\sqrt{\frac{(\boldsymbol{y} - \boldsymbol{u}(0))^T\boldsymbol{H}(0)^{-1}(\boldsymbol{y} - \boldsymbol{u}(0))}{N}}\right) + \sqrt{\frac{\log(1/\delta)}{N}},
\end{aligned}$$

where $(i)$ follows from $\tilde{f}_\infty(\boldsymbol{X}) = \boldsymbol{y}$ and we obtain $\hat{R}_S(\tilde{f}_\infty) = 0$.

