# OpenReview forum: "On the optimization and generalization of overparameterized implicit neural networks"
_ICLR.cc/2023/Conference — Submitted to ICLR 2023_

### Official Review · Reviewer_EsXR · 2022-10-25

**Confidence:** 3
**Correctness:** 3
**Technical Novelty And Significance:** 3
**Empirical Novelty And Significance:** Not applicable
**Recommendation:** 5

**Clarity, Quality, Novelty And Reproducibility:**

As mentioned in the Strength And Weaknesses section, I think the clarity of the paper needs improvement.

In terms of novelty, a large of the proof in the paper is to mimic the standard NTK analysis, so the novelty mainly lies in the adjustments to the implicit neural network setting.

**Strength And Weaknesses:**


The analysis in this paper looks solid and interesting. The results are also important and interesting. However, I also have the following concerns.

1. To my knowledge, the properties of the neural tangent kernel for implicit neural networks have not been thoroughly studied. Therefore, it is difficult to judge the actual impact of the results in this paper. For example, the main results in this paper all rely on the assumption that $\lambda_{\min}(\mathbf{H}(0)) \geq \lambda_0 > 0$ with probability 1. While this assumption is likely correct it needs to be proved. Moreover, in

Hayou, Soufiane, Arnaud Doucet, and Judith Rousseau. "Exact Convergence Rates of the Neural Tangent Kernel in the Large Depth Limit." arXiv preprint arXiv:1905.13654 (2019),

it is proved that the neural tangent kernel is degenerate in the infinite depth limit. Although I understand that implicit neural networks are different from standard infinitely deep neural networks, whether the NTK is meaningful still requires rigorous demonstration.

2. The presentation of the paper is not clear and there are many typos. For example, in the first paragraph:

dominated -> dominant
less memory resources -> fewer memory resources
“weight-tied” and “input-injected” are adjectives

In the theorems, it seems that $\mathbf{A}(t)$ should be $\| \mathbf{A}(t) \|$. Moreover, the citation format looks a bit weird. Perhaps the authors misused the “\citet” and “\citep” commends in natbib. The organization of the paper may also be improved if the authors move proof sketches to the later part of the paper.

**Summary Of The Paper:**

This paper studies the learning of over-parameterized implicit neural networks in the neural tangent kernel regime. The authors establish convergence guarantee of gradient flow, and also provide a generalization bound for the obtained implicit neural network.

**Summary Of The Review:**

Based on the concerns mentioned in the Strength And Weaknesses section, I think this paper needs a thorough revision before publication.

---

### Official Review · Reviewer_vuH7 · 2022-10-25

**Confidence:** 4
**Correctness:** 4
**Technical Novelty And Significance:** 2
**Empirical Novelty And Significance:** 2
**Recommendation:** 5

**Clarity, Quality, Novelty And Reproducibility:**

Clarity and quality are good. The novelty seems to be a bit limited.


**Strength And Weaknesses:**

Strength:

* This paper proves the global convergence of gradient flow for training implicit neural networks. Compared with prior works, this paper considers only training the implicit layer, which is more challenging.
* The authors provide an initialization-sensitive generalization bound.


weakness:
* Though the authors claim that studying training over implicit layers is more challenging compared to training over (almost) the last layer.  theoretical analysis are still very similar. Besides, there are nearly no new findings one can gain from the developed results.
* The so-called initialization-sensitive generalization bound is not a big contribution. Almost all NTK based generalization analysis can give an initialization-sensitive generalization bound, as long as the optimization trajectory is sufficiently close to the initialization.
* Deeper explorations are needed toward understanding and advancing the initialization of neural networks. It would be better to provide a quantitative bound on the smallest eigenvalue of $H(0)$ and discuss whether there are better initialization methods to improve this quantity.


**Summary Of The Paper:**

This paper studies the optimization and generalization of learning over-parameterized implicit neural networks, where only hidden layers are trained. The authors proved global convergence of gradient descent and provide a generalization bound that is initialization sensitive.


**Summary Of The Review:**

Overall, this paper is theoretical sound. My major concern lies in the contribution of this paper. It seems that the exploration is not deep enough given prior works.

---

### Official Review · Reviewer_7aUe · 2022-10-26

**Confidence:** 2
**Correctness:** 4
**Technical Novelty And Significance:** 3
**Empirical Novelty And Significance:** 1
**Recommendation:** 6

**Clarity, Quality, Novelty And Reproducibility:**

**Clarity and Quality:** While some parts of the proofs are a bit opaque, the overall proof strategies are outlined well throughout the paper. There are also frequent grammatical errors throughout the paper (this was not considered when reaching a final conclusion about the paper). I also strongly recommend the authors to fix remaining typos, especially the wrong use of \citet used throughout the submission.

**Novelty:** The initialization-based generalization bound, as well as the convergence result for the implicit layer appear to be novel. (I don't have extensive background in this field, hence might have missed relevant classical or contemporary work)

**Reproducibility:** Since this work is mostly theoretical, there's no major reproducibility concern.

**Strength And Weaknesses:**

**STRENGTHS**
* **Convergence:** The convergence proof addresses an existing gap in the literature. Showing that the implicit layer does converge (without basing the proof on the dynamics of the readout layer) provides a better understanding of the training dynamics.
* **Initialization based generalization bound:** Empirically, it is indeed the case that the way DEQ models are initialized have an impact on the training dynamics and generalization. While not conclusive, the authors' approach to derive an initialization-dependent bound might be useful in understanding what aspect of an initialization drives favourable convergence and generalization. The following two insights look particularly interesting:
  * Aligning the error vector $\mathbf{u}(0) - \mathbf{y})$ with the eigenvectors corresponding to large eigenvalues of $\mathbf{H}$, a particular Gram-matrix like quantity.
  * Ensuring that the eigenvalues of this Gram matrix-like quantity are close to each other also improves generalization.

****

**WEAKNESSES**
* **Lack of empirical corroboration:** It's not clear whether the bound provided in the paper is tight for the regimes under which DEQs are trained. The experiments section of the paper doesn't go further than demonstrating that wider DEQs converge faster to smaller test losses.
* **Analysis focuses on a restrictive class of DEQs:** The analysis only considers a restricted class of DEQ models that are rarely used in practice.
* **Potentially limited novelty:** While the results presented by the authors do fill existing gaps in the literature, it doesn't introduce novel proof techniques while doing so, and appear to be following proof techniques outlined in NTK based convergence and generalization analyses.

****

**QUESTIONS TO AUTHORS:**
* How do you make sure that the forward pass is well-posed during the gradient flow? The current answer appears to be "by picking a $\gamma_0$ small enough to ensure contractivity throughout training". I was wondering if this puts an artificial cap on how "deep" DEQs can operate, as small choices $\gamma_0$ will lead to very quick convergence of the forward pass, effectively making the DEQ effectively shallow.

**Summary Of The Paper:**

The authors study convergence and generalization of implicit models (in particular simple deep equilibrium models).

The submission aims at addressing the following gaps in the literature:
* **Existing convergence results rely on studying the read-out layer:** The authors address this by deriving a convergence result that explicitly focuses on the equilibrium layer.
* **A generalization bounds are initialization-agnostic:** The authors derive a Rademacher complexity-based generalization bound that is initialization sensitive. Their result has useful practical connotations: for example, one can try improving generalization by picking an initialization that 1) aligns the error vector ($\mathbf{u}(0) - \mathbf{y})$ where $\mathbf{u}$ is the output of the equilibrium model at initialization and $\mathbf{y}$ is the label vector) with the eigenvectors corresponding to large eigenvalues of a Hessian-like quantity that can be estimated at initialization).



**Summary Of The Review:**

This submission improves our understanding of the convergence (under gradient flow) and generalization of simple deep equilibrium models. So long as the proofs are correct, it fills a number of existing gaps in the literature.



(Important disclaimer: I cannot attest to the correctness of all the proofs. I've only confirmed the correctness of a few sampled proof segments in the main body of the paper)

---

### Decision · Program_Chairs · 2023-01-20

**Decision:**

Reject

**Justification For Why Not Higher Score:**

See above.

**Justification For Why Not Lower Score:**

N/A

**Metareview: Summary, Strengths And Weaknesses:**

This paper analyzes the convergence and generalization of deep equilibrium models (DEQs), adapting traditional NTK-based analysis to implicit models. Overall, this seems like a worthwhile paper: DEQs are an increasingly relevant model class, and in light of their notorious optimization difficulties, it is useful to have a theoretical analysis of optimization (especially if it can yield insight into initialization). Reviewers expressed various concerns, such as about the clarity (and the paper does appear a little rushed). Reviewers also questioned the novelty (are new techniques required for the implicit setting, or is it mostly a standard NTK analysis?), the reasonableness of assumptions, and the lack of experimental validation. It's likely that the authors could have addressed many of these points in their response, but unfortunately they seem not to have submitted one.